# Considering the Value of 3D Cultures for Enhancing the Understanding of Adhesion, Proliferation, and Osteogenesis on Titanium Dental Implants

**DOI:** 10.3390/biom13071048

**Published:** 2023-06-28

**Authors:** Federico Ferro, Federico Azzolin, Renza Spelat, Lorenzo Bevilacqua, Michele Maglione

**Affiliations:** 1Department of Medical and Biological Sciences, University of Udine, 33100 Udine, Italy; 2Department of Medical, Surgery and Health Sciences, University of Trieste, 34125 Trieste, Italy; federico.azzolin@gmail.com (F.A.); l.bevilacqua@fmc.units.it (L.B.); m.maglione@fmc.units.it (M.M.); 3Neurobiology Sector, International School for Advanced Studies (SISSA), 34136 Trieste, Italy; renza.spelat@libero.it

**Keywords:** cell–surface interactions, combinatorial approach, implantology, surface micro-topography, three-dimensional cell culture

## Abstract

Background: Individuals with pathologic conditions and restorative deficiencies might benefit from a combinatorial approach encompassing stem cells and dental implants; however, due to the various surface textures and coatings, the influence of titanium dental implants on cells exhibits extensive, wide variations. Three-dimensional (3D) cultures of stem cells on whole dental implants are superior in testing implant properties and were used to examine their capabilities thoroughly. Materials and methods: The surface micro-topography of five titanium dental implants manufactured by sandblasting with titanium, aluminum, corundum, or laser sintered and laser machined was compared in this study. After characterization, including particle size distribution and roughness, the adhesion, proliferation, and viability of adipose-derived stem cells (ADSCs) cultured on the whole-body implants were tested at three time points (one to seven days). Finally, the capacity of the implant to induce ADSCs’ spontaneous osteoblastic differentiation was examined at the same time points, assessing the gene expression of collagen type 1 (*coll-I*), osteonectin (*osn*), alkaline phosphatase (*alp*), and osteocalcin (*osc*). Results: Laser-treated (Laser Mach and Laser Sint) implants exhibited the highest adhesion degree; however, limited proliferation was observed, except for Laser Sint implants, while viability differences were seen throughout the three time points, except for Ti Blast implants. Sandblasted surfaces (Al Blast, Cor Blast, and Ti Blast) outpaced the laser-treated ones, inducing higher amounts of *coll*-*I*, *osn*, and *alp*, but not *osc*. Among the sandblasted surfaces, Ti Blast showed moderate roughness and the highest superficial texture density, favoring the most significant spontaneous differentiation relative to all the other implant surfaces. Conclusions: The results indicate that 3D cultures of stem cells on whole-body titanium dental implants is a practical and physiologically appropriate way to test the biological characteristics of the implants, revealing peculiar differences in ADSCs’ adhesion, proliferation, and activity toward osteogenic commitment in the absence of specific osteoinductive cues. In addition, the 3D method would allow researchers to test various implant surfaces more thoroughly. Integrating with preconditioned stem cells would inspire a more substantial combinatorial approach to promote a quicker recovery for patients with restorative impairments.

## 1. Introduction

Bone regeneration is limited in the elderly and individuals with pathologic disorders such as osteoporosis, diabetes, and metabolically linked diseases after trauma or surgery [1,2,3,4]. The need for restorative dental treatments is anticipated to increase as the population ages, and since many pathologies have a negative impact on the success of rehabilitation, being responsible for a significant portion of potential implant failure, it is crucial to improve therapeutic approaches and develop new and more efficient ones [1,2].

The progress in current implantology has depended on the surface functionalization of restorative materials to improve their biocompatibility and in vivo performance [1,5,6]. Therefore, research in the field has centered mainly on various techniques that affect the surface topography of implants, such as changing the surface texture of the implant. The surface modification techniques used to increase microroughness can be obtained by acid etching, sandblasting, heat treatments, anodic oxidation, and the combination of any of these treatments [7,8,9,10]. Indeed, the macro- and micro-topography of oral implants dramatically affect their stability and duration [7,8,9]. Specifically, the charge, geometry, and chemical composition of the surface, in particular, ensure optimal load transfer to the bone and the surrounding tissues, attract differentiated or undifferentiated progenitor cells from the native tissue, and induce regeneration and differentiation of host progenitor cells favoring, in fact, osseointegration and long-term stability of the implant [8,11].

In addition, in pathological settings or in the presence of elderly individuals, it is difficult to increase implant osseointegration using only physical–chemical changes because bone repair and remodeling are complicated processes involving the migration, proliferation, and differentiation of osteogenic cells [1]. Significant progress has been achieved in using material engineering and stem cells to better understand and treat a range of restorative deficiencies or pathologic conditions, and preclinical attempts to combine material engineering with stem cells have also been made [1,2,12,13,14,15]. Indeed, recent advances in combinatorial strategies using stem-cell-based tissue engineering have improved titanium implant osseointegration in diabetic and osteoporotic animal models [1,12,16].

The findings notably showed that dental implants combined with cell-based tissue engineering approaches exhibited increased cell density, extracellular matrix (ECM) content, growth factors, and stability surrounding the bone–implant contact area [17]. Previous studies have also demonstrated that stem cell sheets may be used to create a hybrid cell–implant construct with osteogenic potential in vitro and in vivo [1,18]. According to these investigations, this combinational strategy improves titanium implants’ ability to osseointegrate in healthy, diabetic, and osteoporotic models, potentially expanding their therapeutic use in patients with restorative deficiencies and increasing the success rate of dental implants [1,12,16].

On the downside, the impact of titanium dental implants on cells varies significantly because of the different surface textures and coatings [5,6,10,19,20,21,22,23]. Nevertheless, several previous studies examining the effects of coating and surface roughness on whole-body titanium dental implants established the approach’s feasibility by demonstrating a significant advantage from testing stem cells in 3D in terms of adhesion and differentiation [24,25]. In fact, stem cells grown on implants display an improved differentiation compared to the same cells maintained in 2D because 3D cell cultures better resemble the physiological cell environment [25]. Therefore, despite being a more concrete and physiologically appropriate approach, little is known about the effects of the 3D culture of stem cells on whole-body implants, including attraction, adhesion, and, secondarily, proliferative and differentiation potential.

As a consequence, the present study aims to examine whether the micro-geometry and surface properties of five titanium implants affect the adhesion, proliferation, and differentiation in vitro of a population of adipose-tissue-derived stem cells (ADSCs) in order to strengthen the groundwork for a more realistic combinatorial strategy and confirm the significance of testing cells in a 3D experimental condition.

## 2. Materials and Methods

### 2.1. Cell Culture Products and Reagents

Lipoaspirates were obtained from healthy donors over the age of 18 after obtaining informed consent and following all regulatory requirements for confidentiality and the administration of biological material, according to the ethical committee of the University of Udine (Par. N. 103/2011/Sper). The donors’ lipoaspirate (15 mL) was obtained using the tumescent procedure, which involves infusing Klein’s solution and then recovering the adipose tissue. After that, the adipose tissue was centrifuged at 1800× *g* for 15 min to remove the red blood cells. ADSCs were placed in Joklik modified alpha-MEM medium (Gibco, Carlsbad, CA, USA) supplemented with 400 U/mL collagenase type 2 (Wortington, Columbus, OH, USA) equal to two times the volume of lipoaspirates and incubated for 15 min at 37 °C under moderate shaking every 2–3 min. The solution was then centrifuged for 10 min at 1800× *g*. After that, the supernatant was discarded, and the pellet was resuspended and filtered through a 70 μm filter. ADSCs were cultured onto 100 mm dishes (2 × 10^6^ cells per dish) in the presence of a proliferation medium composed of α-MEM with l-glutamine 2 mmol/L, 10% fetal bovine serum (FBS), insulin 10 mg/mL, dexamethasone 10^−9^ mol/L, ascorbic acid 100 mM, epithelial growth factor 10 ng/mL, and gentamicin 50 ng/mL (all from Sigma-Aldrich, Saint Louis, MO, USA). To inhibit cell differentiation, ADSCs were kept semi-confluent, and 80% of the media was changed every three days. ADSCs were maintained at 37 °C, 5% CO_2_, and a humidified atmosphere in the proliferation medium. A bland detaching solution, namely CTC (collagenase 20 U/mL, trypsin 0.75 mg/mL, 2% heat-inactivated dialyzed chicken serum (Gibco) in calcium, and magnesium Hank’s balanced salt solution (HBSS)), was used to detach ADSCs. Cells between passages 1 and 6 (P1 and P6) were used in the experiments [24,26].

### 2.2. Characteristics of Dental Implants

Five different titanium dental implants, 1: sandblasted with aluminum oxide and acid etched (Al Blast, N = 9); 2: laser-sintered (Laser Sint, N = 9); 3: sandblasted with titanium dioxide spheres and fluoride treatment (Ti Blast, N = 9); 4: laser-machined (Laser Mach, N = 9); 5: sandblasted with corundum and different acid etchings (Cor Blast, N = 9), have been tested and compared throughout this study. The implants had a tronco-conical shape with dimensions ranging from 3.6 to 4 mm in diameter and 8 to 9 mm in length and were chosen for their wider market representation, which could be attributed to their higher difference in roughness degree and surface texture compared to our previous research [24]. The surface of Al Blast, which is made of grade 4 titanium, has a uniform finish due to a temperature-controlled process of large-grit sandblasting with aluminum oxide (Al_2_O_3_, around 0.5 mm) and acid etching. According to the manufacturer, no surface debris is left from the blasting and etching procedure, which creates a uniform surface with a very high protein-binding capability before being sterilized by a plasma cold reactor and packaged under clean, low-germ conditions.

The Laser Sint implant, made of Ti_6_Al_4_V, is created by sintering metal powder nanoparticles with one another and against the implant surface using a selective laser melting technique via a computer-controlled laser beam to produce a complex implant surface. The final texture is an intercommunicating surface that interacts with the host bone and permits osteoblastic cells to enter the implant body. The implant is dried and made ready for use after being cleaned with acetone, ethanol, and distilled water in an ultrasonic cleaner.

The Ti Blast implant is made of Ti_6_Al_4_V alloy and has a moderately rough surface resulting from two subtractive, sequential manufacturing processes. The titanium dioxide sandblasting spheres (0.25–0.75 mm) give a microscale surface roughness, and the subsequent etching with hydrofluoric acid shapes the nanostructure of the implant. The implant is dried and made ready for use after being sterilized and packaged in an unspecified solution.

The Laser Mach implant, made of Ti_6_Al_4_V alloy, has a sequence of finely carved, cell-sized networks manufactured through a laser subtractive process using a high-energy and highly concentrated laser beam from a CO_2_ laser source ranging from 10 W up to 30 W. The implant is dried, sterilized by γ irradiation, then packed and made ready for use.

The Cor Blast implant, made of extra-low interstitial Ti_6_Al_4_V alloy, is etched with different acids (HCl and H_2_SO_4_) at high temperatures and then sandblasted with corundum (Al_2_O_3_, a naturally occurring mineral of aluminum oxide, 0.25–0.5 mm) at high pressure (up to 5 bar) to provide a highly homogeneous surface. The implant has a high level of cleanliness, which is due to the decontamination procedure in a cold plasma reactor.

### 2.3. Dental Implants Surface Characterization

Morphological analyses were carried out using a scanning electron microscope, Quanta250 (SEM; FEI, Hillsboro, OR, USA), in high vacuum and secondary electron mode with a tension of 30 kV; the operational distance was set in order to obtain the proper magnification. Aluminum stubs covered with carbon double-sided tape were used to secure the samples. The samples were then gold sputtered with a Sputter Coater K550X (Emitech; Quorum Technologies Ltd., Laoughton, UK).

The implants were then subjected to acquisitions at higher magnifications (40, 100, 800, and 1500×), and three principal profiles were obtained for each implant perpendicular to the implant longitudinal axis. To evaluate the surface micro-morphology, average size, distance between the particles, and number of particles, SEM photomicrographs (N = 3 per implant surface, magnification 1500×, considering an area of 100 × 100 µm) were taken between two threads and, after binarization and thresholding, were analyzed using ImageJ and the nearest distances (ND) plugin (http://imagej.nih.gov, USA, accessed on 3 March 2023) [27].

Although initially created for different purposes and highlighting the measurement uncertainty and noise that can be caused by errors induced by this method and by the roughness estimation process itself [28,29,30], the plugin SurfCharJ was used for approximately determining the roughness parameters from the SEM microphotographs [20]. Therefore, roughness was assessed according to ISO 4287/2000, measuring the following parameters: Ra (arithmetical mean deviation), Rq (root mean square deviation), Rsk (skewness of the assessed profile), and Rku (kurtosis of the assessed profile) using SEM photomicrographs and the SurfCharJ plugin [20,24]. Ra and Rq specifically describe the topological events’ height and amplitude (peaks and valleys). In contrast, Rsk and Rku describe the distribution and are used to specify both the symmetry and the tailness of the peaks and valleys [10]. The plot profiles of the different implants were obtained using the ImageJ plugins, the surface plot profile, and the 3D surface plot [21,24].

### 2.4. Dental Implant Seeding with ADSCs

Each dental implant (N = 9) was placed vertically in a 96-multiwell plate (Gibco), which was contained in a 150 mm plate to maintain sterility. ADSCs between passages 3 and 6, kept semi-confluent to avoid spontaneous differentiation and maintained at 37 °C with 5% CO_2_ in a humidified atmosphere in the proliferation medium, were washed in PBS (Sigma), detached using CTC, and centrifuged for 5 min at 1800× *g*. Following this procedure, ADSCs were resuspended in the proliferation medium to a final concentration of 6.25 × 10^6^/mL and 200 µL of medium was added at time point zero (T0) to each well. ADSCs were allowed to adhere to the implants, and after 24 h (T1), non-adherent cells were removed, transferring the implants into a clean 96-multiwell plate (Gibco) filled with fresh proliferation medium (200 µL) inside a 150mm plate. On day three (T2), the proliferation medium was changed as described for day one and left at 37 °C with 5% CO_2_ in a humidified atmosphere until day seven (T3) [24].

### 2.5. Cell Adhesion, Viability and Proliferation

The procedure for cell detachment from the implants involved transferring the implants into 1.5 mL conical tubes containing CTC, followed by 10 min of incubation at 37 °C in a humid environment with 5% CO_2_. Cells were then centrifuged for 5 min at 1800× *g* and counted in triplicate at each time point (T1–T3) in the Neubauer chamber. ADSCs’ adhesion, viability, and proliferation were evaluated in triplicate (N = 3) starting from T0 and throughout the three time points (T1–T3). Adhesion was considered as the number of adhering cells per mm^2^ across the time point of interest and the other time points on the respective implants. Viability was considered as the ratio of dead cells across the time point of interest and the other time points on the respective implants. Proliferation was considered as the difference in cells counted across the time point of interest and the other time points on the respective implants. The total number of cells seeded at T0, as well as the surface of the submerged implant, were used to normalize the adhesion, viability, and proliferation assessment results [24,26].

### 2.6. Gene Expression Analysis

After being washed in PBS (Sigma), detached using CTC, and centrifuged for 5 min at 1800× *g*, total RNA was extracted from ADSCs cultured on each different implant (N = 3) using TRIzol (Gibco). RNA samples were then quantified by spectrophotometer and treated with DNase I (Ambion, Carlsbad, CA, USA). First strand cDNA synthesis was performed with 100 ng of total RNA using M-MLV reverse transcriptase (Invitrogen). PCR amplification was performed in a final volume of 50 µL, using 10 ng of cDNA, 50 pmol of each primer, and 2 U/µL of TaqI polymerase (Amersham, Piscataway, NJ, USA). The following primers were employed to quantify the expression of osteoblastic markers on days one, three, and seven in triplicate: alkaline phosphatase (*alp*) GCAGGCAGGCAGCTTCAC, TCAGAACAGGACGCTCAGG, 496 bp, 60.5 °C, NM_000478.3; collagen Type I (*coll*-*I*) TAAAGGGTCACCGTGGCT, CGAACCACATTGGCATCA, 355 bp, 60 °C, NM_000088.3; osteocalcin (*osc*) TCACACTCCTCGCCCTATTG, CTAGACCGGGCCGTAGAAG, 293 bp, 58 °C, NM_199173.3; and osteonectin (*osn*) CACAAGCTCCACCTGGACTA, GAATCCGGTACTGTGGAAGG, 525 bp, 58 °C, NM_003118.2. RNA polymerase type II GCACCACGTACACCAATG, GTGGGGCTGCTTTAACCA 350 bp 56 °C NM_000937 was used as a housekeeping gene (MWG Eurofins, Ebersberg, DE) [24,26].

### 2.7. Statistical Analysis

The data are presented as mean ± standard deviations (SD) for cell counts and gene expression, implant roughness, and surface particle characteristics. The statistical significance was ascertained using the analysis of variance (ANOVA) and Bonferroni’s post hoc tests. A paired *t*-test was used to compare data from similar groups. Statistical significance was determined by utilizing SPSS 23.0 (SPSS IBM, USA) to analyze the data, and significance is indicated as + Al Blast, * Laser Sint, £ Ti Blast, % Laser Mach, and ^ Al Blast (ANOVA followed by Bonferroni’s post hoc test), and * T1 and 24 h–3 days, ^ T2 and 24 h–7 days, and % T3 and 3–7days (paired *t*-test) with *p* ≤ 0.05 or *p* ≤ 0.001.

## 3. Results

### 3.1. ADSCs’ Isolation, Characterization, and Stemness Potential Assessment

ADSCs were isolated from lipoaspirates taken from healthy donors using the tumescent technique as described in our previously published works [24,26].

In summary, flow cytometry was used to show the stemness-related membrane proteins in ADSCs at passage three (P3), demonstrating the presence of CD73, CD90, and CD105, but not the hematopoietic CD34, CD45, and CD106, as previously reported [24]. To validate their multi-lineage capability, ADSCs were then differentiated through the adipocyte, osteoblastic, and chondroblastic lineages. Adipocyte induction was confirmed by the presence of lipid droplets detected using Oil Red O staining and compared with non-induced cells. Osteoblastically differentiated ADSCs differed from undifferentiated cells by displaying a high quantity of calcium, identified using alizarin red. Finally, Safranin O staining was performed to validate and compare the chondroblastic differentiation of induced ADSCs to that of control cells [24,26].

### 3.2. Implants’ Morphological Characterization

SEM microphotographs of the implant surfaces revealed their characteristic morphology (Figure 1). The texture of the Al Blast implant showed a high level of regular roughness and the presence of numerous deep cavities. Considering that pits and holes are easily invaded by bone cells, the geometry and connectivity of these cavities may allow bone cells to infiltrate the implant body deeply, allowing for better integration (Figure 1A). Small irregularities with a shattered look were more evident on the Cor Blast implant (Figure 1B). The Laser Sint implant showed the highest heterogeneity with either a sharp or round morphology (Figure 1C). Less frequent, but larger, irregularities with a sharp look were more evident on the Laser Mach and Ti Blast implants (Figure 1D,E).

The microarchitecture of a biomaterial, including the number of particles and their size, distribution, and density, dramatically affects its interaction with tissues and cells and influences the mechanical properties of the scaffold itself [27]. In particular, it has been observed that the texture of the implant surface significantly affects cells both in vitro and in vivo, encouraging osseointegration [7,8,9,27] and likely extending its duration and long-term stability [8,11]. To establish whether or not particles were evenly distributed throughout the surface and would have an impact on cell adhesion and proliferation, we thus analyzed the surface mean particle area, distance, spacing, and number (Figure 2A–J, Table 1).

The results demonstrated no significant difference in the mean area of the surface particles between the tested implants (Figure 2A–F; Table 1) (*p* > 0.05). The Ti Blast implant displayed a slightly lower mean distance between particles (4.06 ± 0.3 μm) than the Laser Mach (5.4 ± 0.76 μm) and Cor Blast (6.17 ± 1.17 μm) implants (*p* ≤ 0.05) (Figure 2A–E,G; Table 1).

The nearest particle distance of the Al Blast (4.26 ± 0.38 μm), Laser Mach (4.34 ± 0.49 μm), Cor Blast (4.86 ± 0.69 μm) and Laser Sint (4.07 ± 0.2 μm) implants was similar, while in the Ti Blast implant (3.5 ± 0.22 μm), the distance was significantly smaller than the rest (*p* ≤ 0.05) (Figure 2A–E,H; Table 1).

The average wall thickness between the particles of Ti Blast (1.44 ± 0.25 μm) was the smallest and was significantly lower than in the Cor Blast (2.93 ± 1.21 μm) and Laser Mach (2.24 ± 0.74 μm) implants, but the difference was not significant with respect to the Al Blast (2.12 ± 0.76 μm) or Laser Sint (1.98 ± 0.3 μm) implants (*p* ≤ 0.05) (Figure 2A–E,I; Table 1). The results indicated that the surfaces with the greatest and lowest number of surface particles were, respectively, the Ti Blast (506 ± 18) and Cor Blast (174 ± 69 part/mm^2^) implants (*p* ≤ 0.05). The number of surface particles on the Al Blast (250 ± 1.5), Laser Sint (304 ± 39), and Cor Blast (174 ± 69) implants were different from each other (*p* ≤ 0.05) but no significant differences were seen with respect to the Laser Mach implant (273 ± 126) (Figure 2J, Table 1).

Surface roughness of titanium implants, particularly moderate roughness, allows for improved cell and bone connections [31,32,33] and mastication load transfer, which also promotes more osteoblastic and progenitor cell differentiation [8,34,35] than smoother or rougher surfaces. Consequently, according to ISO 4287/2000, a rough estimation of roughness was considered using the ImageJ plugin SurfCharJ to measure the following surface roughness parameters: Ra, Rq, Rsk, and Rku [20,24]. The 3D plots (Figure 3A–E) and surface roughness profiles of the implants (Figure 3F–J) were created using the Al Blast, Laser Sint, Ti Blast, Laser Mach, and Cor Blast SEM microphotographs, which were then merged in Figure 3K to emphasize their differences.

Following analysis of the arithmetical mean deviation (Ra), root mean square deviation (Rq), skewness of the assessed profiles (Rsk), and kurtosis of the assessed profiles (Rku) [20], substantial differences were evidenced and expressed as the mean ± SD of three independent experiments, with *p* ≤ 0.05 indicating significance (ANOVA followed by Bonferroni’s post hoc test) (Figure 4A–D; Table 2).

In comparison to all surfaces, Al Blast and Cor Blast implants had the greatest values for Ra (1.64 ± 0.07 μm and 1.64 ± 0.21 μm) and Rq (2.04 ± 0.04 μm and 2.04 ± 0.1 μm), respectively (all *p* ≤ 0.05) (Figure 4A,B; Table 2). In contrast, the Laser Mach implant had the lowest values for Ra (0.8 ± 0.1 μm) and Rq (1.42 ± 0.1 μm) compared to the other implant surfaces (*p* ≤ 0.05) (Figure 4A,B; Table 2).

The roughness-associated parameters Rsk and Rku for the Laser Mach implant were the highest at 1.79 ± 0.15 μm and 3.31 ± 0.56 μm, respectively, and differed considerably from all other implants (*p* ≤ 0.05) (Figure 4D,E; Table 2), whereas the Al Blast implant exhibited the lowest values: 1.23 ± 0.04 μm and 1.56 ± 0.06 μm (*p* ≤ 0.05) (Figure 4C,D; Table 2).

The surface texture and roughness analyses revealed that the Ti Blast implant had the highest particle density (comprehensive of the highest number of particles, the smallest nearest particle distance, and a lower trend for the mean distance between particles and average wall thickness), even though there were no appreciable differences in their area. Instead, as seen by the corresponding indicators (Ra and Rq), the Al Blast and Cor Blast implants exhibit the greatest degree of roughness, whereas the Laser Mach implant had the lowest values. As expected, the Laser Mach implant also displayed the highest Rsk and Rku values, as opposed to the Al Blast implant.

### 3.3. ADSCs’ Adhesion

In order to address any issues related to the seeding strategy, the ratios of adherent cells retrieved at the three time periods were standardized according to the total number of cells at day zero (1.25 × 10^6^ cells) and the surface area of the implants (Figure 5A–D). The differences emphasized by the data are shown as the mean ± SD of three separate experiments, with *p* ≤ 0.001 indicating significance (ANOVA followed by Bonferroni’s post hoc test and paired *t*-test).

On day one, the highest adhesion was found for the Laser Mach implant (159 ± 32.8 cells/mm^2^), followed by the Laser Sint (152.8 ± 36.4 cells/mm^2^), Ti Blast (87.1 ± 20.1 cells/mm^2^), Al Blast (65.3 ± 30 cells/mm^2^), and lastly, Cor Blast (59.2 ± 48 cells/mm^2^) implants (Figure 5A; Appendix A) (*p* ≤ 0.001). The Cor blast, Al Blast, and Ti Blast implants exhibited the lowest adhesion with respect to the remaining surfaces (*p* ≤ 0.001) (Figure 5A; Appendix A).

By day three, we discovered an increase in the number of cells with respect to day one only on the Cor Blast implant (145 ± 71.7 cells/mm^2^) (*p* ≤ 0.001) (Figure 5B, Appendix A), whereas the number decreased on the Laser Mach implant (108.5 ± 40 cells/mm^2^) (*p* ≤ 0.001) or remained stable on the other implants, including the Laser Sint (127.4 ± 62.1 cells/mm^2^), Al Blast (67.3 ± 37.6 cells/mm^2^), and Ti Blast (79.2 ± 56.2 cells/mm^2^) implants (Figure 5A, Appendix A).

At the three-day time point, the number of cells adhering to the Cor Blast implant was significantly different from those on the Ti Blast (*p* = 0.001) and Al Blast (*p* ≤ 0.001) implants; similarly, the Laser Sint implant differed from the Ti Blast (*p* = 0.032) and Al Blast (*p* = 0.002) implants. The Ti Blast and Al Blast implants differed from the Laser Sint and Cor Blast implants (*p* = 0.001 and *p* ≤ 0.001, respectively) (Figure 5D and Appendix A).

Between days three and seven, we discovered a proliferative phase on Laser Sint (199.6 ± 63.2 cells/mm^2^) (*p* = 0.008) implants, whereas on Laser Mach (138.2 ± 83 cells/mm^2^) implants, the increase was not significant. On Al Blast (35.7 ± 16.2 cells/mm^2^) implants, the cell number exhibited a significant reduction (*p* ≤ 0.001), while the reduction was not significant on Cor Blast (114 ± 64.2 cells/mm^2^) and Ti Blast (62.7 ± 40.6 cells/mm^2^) implants (Figure 5A; Appendix A).

On the seventh day, the cell number on the Ti Blast implant varied considerably from the Laser Mach (*p* ≤ 0.001), Laser Sint (*p* ≤ 0.001), and Cor Blast (*p* = 0.024) implants (Figure 5F and Appendix A). The Al Blast surface differed from the Laser Mach, Laser Sint and the Cor Blast surface (all *p* ≤ 0.001). The Laser Mach implant showed adhesion differences with the Ti Blast (*p* ≤ 0.001), Al Blast (*p* ≤ 0.001) and Laser Sint (*p* = 0.003) implants, but not with the Cor Blast implant (Figure 5B–D; Appendix A).

The Laser Sint implant differed from the Ti Blast, Al Blast, and Cor Blast implants (all *p* ≤ 0.001), and from the Laser Mach implant (*p* = 0.003). The adhesion on the Cor Blast implant was considerably different from those on the Al Blast, Laser Sint (both *p* ≤ 0.001), and Ti Blast (*p* = 0.024) implants (Figure 5B–D; Appendix A).

Finally, comparing the adhesion between the 24 h and seven days time points revealed that the number of cells did not differ significantly for the Laser Mach implant, but did for all the other implants, with a decrease on the Ti Blast (*p* = 0.011), Al Blast (*p* ≤ 0.001), and Cor Blast (*p* = 0.006) implants, and an increase on the Laser Sint implant (*p* = 0.008) (Figure 5B–D; Appendix A).

The cell viability at the 24 h time point with respect to the different implant types demonstrated that the Cor Blast implant (1.7 ± 1.6) revealed better viability with respect to the Al Blast (0.03 ± 0.56), Laser Mach (−0.34 ± 0.37), Laser Sint (−0.22 ± 0.62) and Ti Blast (−0.085 ± 0.69) implants (all *p* ≤ 0.05) (Figure 5E; Appendix A). The results at day three highlighted that the cell viability on the Laser Sint implant (0.78 ± 0.87) was significantly higher than in the Al Blast (−0.5 ± 0.53), Cor Blast (−0.25 ± 0.68), and Ti Blast (−0.19 ± 0.87) implants (all *p* ≤ 0.001), but not in the Laser Mach implant (0.29 ± 0.94) (Figure 5F; Appendix A); meanwhile, the viability of the Al Blast implant was significantly lower than the Laser Mach and Laser Sint implants (*p* = 0.008 and *p* ≤ 0.001, respectively). Lastly, at day seven, the viability of the ADSCs cultured on the Laser Sint (0.32 ± 0.54) and Cor Blast (0.57 ± 0.92) implants was significantly different from those on the Al Blast (−0.88 ± 0.83) and Ti Blast (−0.36 ± 0.64) implants (*p* ≤ 0.001, *p* = 0.011, and both *p* = 0.001, respectively), but not from those on the Laser Mach implant (−0.16 ± 0.55) (*p* > 0.05) (Figure 5G; Appendix A).

Considering the differences in cell viability on the same implants at different time points, the cells on the Al Blast implant showed a gradual decrease through all the time points (*p* = 0.016, *p* = 0.026, and *p* ≤ 0.001), while this decrease was noticeable on the Cor Blast implant only at three days (*p* ≤ 0.001) and on the Laser Mach implant only at 24 h (*p* ≤ 0.019) (Figure 5H; Appendix A). The cell viability on the Laser Sint implant was significant throughout all the time points, reaching a peak at three days, with a *p* ≤ 0.001 at 24 h, *p* = 0.003 at three days, and *p* ≤ 0.001 at seven days (Figure 5H; Appendix A). Finally, the Ti Blast implant showed no discernible fluctuation in cell viability over any given period of time (*p* > 0.05) (Figure 5H; Appendix A).

In summary, a distinctive difference in adhesion was highlighted for laser treated implants (Laser Mach and Laser Sint), which exhibited the highest degrees; however, limited proliferation was observed, specifically for the Laser Sint implant, and viability differences, except for with the Ti Blast implant, were seen throughout the three time points.

### 3.4. Osteoblastic Induction

The efficiency of the experimental seeding approach and the osteoblastic inductive capability of the implant surfaces were assessed by quantifying the expression of ECM and osteoblastic markers and normalizing their values for adherent cell number and surface area. Values are expressed as the mean ± SD of three independent experiments, with *p* ≤ 0.05 indicating significance (ANOVA followed by Bonferroni’s post hoc test). By enhancing the expression of *osn*, *coll*-*I*, and *alp* in comparison to control cells, all of the treatments and surfaces of the whole-body implants were able to promote osteoblastic differentiation, as shown by the results. The Al Blast (152.8 ± 5.6 fold change (FC)), Ti Blast (120.1 ± 4.9 FC), Cor Blast (106.5 ± 7 FC), Laser Mach (66.2 ± 0.73 FC), and Laser Sint (62.9 ± 2.9 FC) implants triggered the highest *coll*-*I* expressions when compared to the control at the 24 h time point (*p* ≤ 0.05) (Figure 6A; Appendix A).

The *osn* expression at 24 h reached the highest value on the Laser Mach implant (14.4 ± 0.4 FC), followed by the Cor Blast (12 ± 0.3 FC) and Laser Sint (7.4 ± 0.3 FC) implants (*p* ≤ 0.05), while on the Ti Blast and Al Blast implants it was not detected (Figure 6C; Appendix A). Additionally, *osc* was not present in all the samples at 24 h and at three days.

Continuing, at three days, the Ti Blast and Al Blast surfaces elicited the greatest *coll*-*I* (respectively, 121.5 ± 2.5 FC and 110 ± 4.2 FC) (*p* ≤ 0.05) and *osn* (respectively, 32.8 ± 0.8 FC and 17.7 ± 0.9 FC) expressions when compared to the other implant types at the same time point (*p* ≤ 0.05) (Figure 6A,C; Appendix A).

At the same time point, the Al Blast and Ti Blast implants expressed the highest quantities of *alp* messenger (respectively, 38.7 ± 0.8 FC and 45.8 ± 0.7 FC) compared to the Laser Sint (24.9 ± 0.25 FC), Laser Mach (11.4 ± 0.3 FC), and Cor Blast (22.9 ± 1.5 FC) implants (*p* ≤ 0.05) (Figure 6B; Appendix A).

By day seven, the Ti Blast and Al Blast treated surfaces still elicited the highest levels of *coll*-*I* (respectively, 165.3 ± 2.9 FC and 215.7 ± 6.2 FC) (*p* ≤ 0.05) and *osn* (respectively, 70 ± 1.6 FC and 82.3 ± 1.5 FC) (*p* ≤ 0.05) (Figure 6A,C; Appendix A) compared to the Laser Mach (respectively, 56.5 ± 0.8 FC and 17.8 ± 0.7 FC), Laser Sint (56.3 ± 3.3 FC and 26.9 ± 0.2 FC), and Cor Blast (104.2 ± 4 FC and 45.9 ± 1.9 FC) implants (*p* ≤ 0.05) (Figure 6A,C; Appendix A).

The *alp* gene was not induced by the Al Blast implant, as at 24 h, but it reached the highest expression peak under the stimulation of the Ti Blast (78.4 ± 2.6 FC) surface, which was significant compared to the Laser Sint (22.1 ± 0.8 FC), Laser Mach (19.3 ± 0.5 FC), and Cor Blast (41.3 ± 0.7 FC) implants (*p* ≤ 0.05) (Figure 6B; Appendix A).

It is interesting to note that by day seven, the late osteoblastic differentiation marker *osc* could not be induced by any surface.

When comparing the expression of the osteoblastic markers with respect to the different time points across the same implant type, it was found that even though the Al Blast implant did not significantly favor *alp* synthesis at 24 h and three days, it was able to induce its expression at seven days (Figure 7A,D; Appendix A). Moreover, despite both Al Blast and Ti Blast implants not inducing the expression of *osn* at 24 h, they were able to significantly induce an increase between day three and day seven (*p* ≤ 0.05) (Figure 7C,F; Appendix A). In contrast, *alp* expression was considerably different between all the intervals (*p* ≤ 0.05). The Cor Blast implant showed significantly increasing *alp*, *osn*, and *coll*-*I* at the 24 h–three day and three–seven day intervals, but was not able to induce a substantial increase in *alp* and *coll*-*I* during the 24 h–three day interval (Figure 7A–F; Appendix A). On the contrary, the production of the *osn* messenger was always positively and significantly influenced by the Cor Blast surface (*p* ≤ 0.05) (Figure 7C,F; Appendix A).

The Laser Mach implant was able to induce significant amounts of messengers for all the tested markers through all the intervals (*p* ≤ 0.05) (Figure 7A–F; Appendix A). The Laser Sint implant successfully induced *osn* through all the intervals, 24 h–three days, 24 h–seven days, and three–seven days (*p* ≤ 0.05), but was able to be inductive for *alp* only during the three–seven day interval (*p* ≤ 0.05) and *coll*-*I* at 24 h–three days (*p* ≤ 0.05) and three–seven days (*p* ≤ 0.05) (Figure 7A–F; Appendix A).

Overall, the results showed that sandblasted surfaces (Al Blast, Cor Blast, and Ti Blast) outperformed the laser-treated ones, promoting the highest spontaneous osteoblastic induction and synthesizing the highest levels of *coll*-*I*, *osn*, and *alp* messengers, although not *osc*. Significantly, among the sandblasted surfaces, the Ti Blast surface promoted, in general, the strongest expression of *coll*-*I*, *osn*, and *alp*.

## 4. Discussion

Since titanium dental implants have a wide range of surface textures and coatings, their impact on cells varies significantly. However, since 3D cell cultures on whole dental implants more closely resemble and mimic physiological cell interactions, they have been proven advantageous in assessing implant capabilities [24,25]. As a fair representation of the micro-morphological roughness of the implant market, we looked at five distinct commercially available dental implants with moderately rough surfaces, testing their potential under 3D conditions.

We found that while roughness is still significant, other factors sensibly influence the adhesive potential of the surfaces of whole-body titanium dental implants. Specifically, the range of surface roughness between Ra 0.8 and 1.2 µm and Rq 1.4 and 1.8 µm does have a beneficial impact on the adhesion of ADSCs.

Implant surfaces within this range of roughness and surface particle texture, such as those found on the Laser Sint and Laser Mach implants, favor a reasonable degree of adhesion compared to the other tested implants, similar to what we previously demonstrated [24]. We also found that Laser Mach and Laser Sint implants had a modest or negligible proliferative phase following the initial adhesion. This observation is also consistent with our previous finding [24] and the cells cultured on the Laser Sint and Cor Blast implants showed the highest viability.

Several studies have demonstrated that the surface characteristics of the dental implants are impacted by manufacturing procedures, changing their composition and affecting the function of tissue and cells [33,36]. Indeed, every process that physically alters the surface of the dental implants also alters their chemistry by introducing new atoms or molecules, which modify protein adsorption and cell adhesion [33,36,37,38]. It is well known that the development of cytotoxicity and inflammation may result from the buildup of solubilized metallic ions on the surface of sandblasted implants [39,40]. A substantial volume of data suggests that throughout the manufacturing or handling process, impurities, whether metallic or nonmetallic, such as F, C, Mg, Fe, Al, Ca, P, Sr, and F, are purposefully or inexorably (despite stringent control) introduced or retained onto the implant surface [33,36,38,41]. We also previously demonstrated that, in contrast to machined and laser micro-patterned treatments, “sandblasting” is more likely to introduce elemental traces of C, Fe, Al, and O [42].

According to these findings, the implants that had undergone laser-related treatments, such as the Laser Sint implant, and to a lesser extent the Laser Mach implant, as well as a laser-treated implant from a previous study [24], appeared to favor ADSCs’ adhesion better, supporting some proliferation as well, or at least were not involved in reduced viability, possibly as a result of higher cleanliness obtained during manufacturing.

The differences in the surfaces and textures showed an influence on ADSCs’ spontaneous differentiation; in this case, the Ti Blast implant, but secondarily the Al Blast implant with some deficiencies, demonstrated the highest inductive potential on the osteoblastic markers *coll*-*I* and *osn*. However, in general, sandblasted implant surfaces, such as the Ti Blast, Al Blast, and Cor Blast surfaces, were more osteoinductive than laser-treated ones. Interestingly, the sandblasted surfaces, Al Blast, Cor Blast, and Ti Blast, underwent treatments involving aluminum, corundum, and fluoride, respectively. Aluminum oxide and corundum (the naturally occurring mineral form of Al_2_O_3_) are deposited during the sandblasting process and seem to have differential effects according to their concentrations. In fact, it was pointed out that high concentrations of residual Al reduce mineralization and osseointegration, while lower concentrations [43,44] have positive effects on cell viability [36] and may be advantageous for bone formation and mineralization [45] and stem cell differentiation [44]. Fluorine surface treatment has been found to clearly affect actin filaments and cell morphology [38,46] and influence cell differentiation, but not proliferation [38,47,48,49].

The findings also showed that the Ti Blast Implant, created by sandblasting with titanium dioxide spheres and etching with hydrofluoric acid, showed no significant effects on adhesion and viability despite its moderate roughness. However, among the other implants, it presented the highest density of surface particles, also suggesting the importance of the texture density.

As an explanation, the presence of this high-density texture may not have been as appealing in terms of cell adhesion, but along with the presence of fluorine ions derived from the surface treatment, it may have been very important in inducing ADSCs’ spontaneous differentiation [19] via the β-integrins, focal adhesion kinase, and Src signaling pathway [50,51]. Indeed, the integrin-mediated interaction between the extracellular matrix proteins, such as fibronectin, vitronectin, Osn, and Coll-I, and the implant surface has been documented to regulate cell adhesion, differentiation, and survival [23,50,51,52].

Previous effects become less consistent when considering that, from our previous results, the implant manufactured through hydroxyapatite and bland acid etching (HA Blasted) performed considerably better than all other implants despite having a similar texture and roughness [24]. Therefore, when surface roughness and manufacturing processes are comparable or identical, the difference in cell adhesion and differentiation between HA Blasted and the other implants might be attributed to the surface treatment (Table 3). Indeed, both the amplitude of surface roughness and the surface chemistry of HA-coated implants have been shown to increase cell adhesion [22,24,52] and subsequent differentiation [22,24,52,53,54,55,56]. In fact, the greater wettability and hydrophilicity stimulate the adsorption of adhesion molecules from the media and, thereby, from the blood onto the titanium implant surface, promoting cell adhesion [52,53,54,55]. Moreover, the HA (Ca_10_(PO_4_)_6_(OH)_2_) coating’s compositional and structural similarity to bone [57] enhances stem cell differentiation [24,57], stimulating extracellular matrix protein secretion and mineralization [56] via integrin-mediated signal transduction [52,58]. On the down side, the primary drawback of using HA is its limited ability to adhere to metal substrates; however, many researchers are making efforts to improve the techniques [57,59,60].

## 5. Conclusions

This study’s results align with previous findings, demonstrating that stem cells are actively influenced by the topography and surface treatment of whole-body implants and the 3D culture approach is capable of unveiling the potential of both [24,25]. Indeed, the results revealed that, in addition to the roughness of the implants under consideration, which were representative of the implant market, the 3D culture approach highlighted substantial differences, in fact:The adhesion and proliferation of the laser-treated implants exhibited the highest degree, possibly due to the influence of the post-processing methods.The sandblasted implant surfaces performed better in terms of osteoinduction, possibly via the chemical treatments the implants underwent during the manufacturing process.The Ti Blast implant’s osteoinductive capability might have benefited from the highest texture density and the fluorine surface treatment.

In light of these results, it is therefore recommended to pursue further research using the 3D culture methodology to broaden and better understand the effects and differences of implant surfaces’ antimicrobial activity, wettability, coatings, or treatments with biologically active molecules, as well as various materials such as zirconia, tantalum, or porcelain.

Finally, by implementing this 3D approach and the subsequent advancement of the combinatorial approach in conjunction with pre-clinical contexts, it would be possible to identify the most efficient and appropriate surface to be used in combination with preconditioned stem cells, promoting a more significant and faster recovery while significantly improving the healing process in individuals with restorative deficiencies.

## Figures and Tables

**Figure 1 biomolecules-13-01048-f001:**
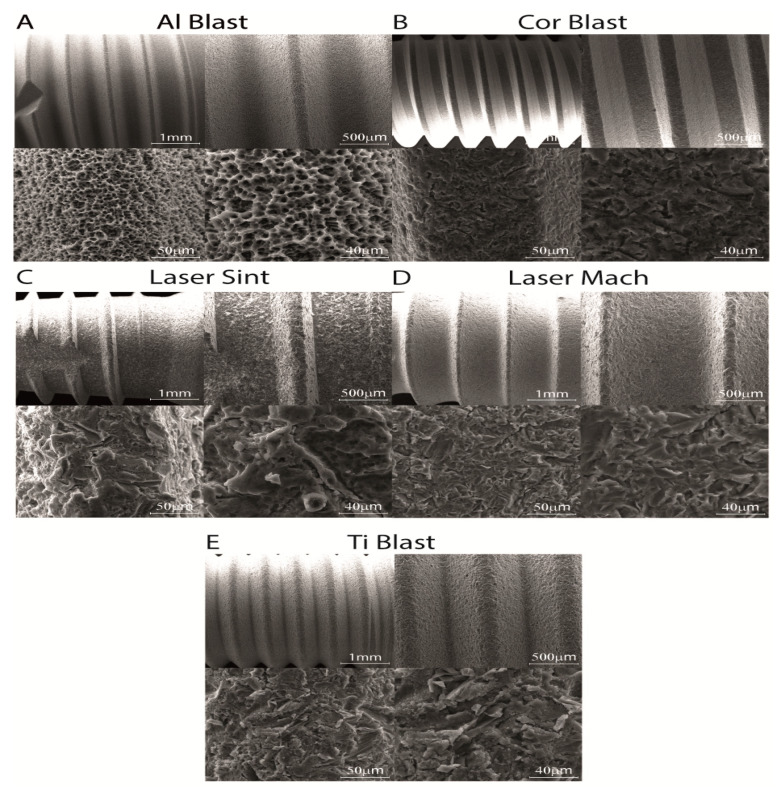
Surface and micro-surface characteristics are presented in SEM photomicrographs. The texture of the different implant surfaces shows the different roughnesses and morphologies of the (**A**) Al Blast, (**B**) Cor Blast, (**C**) Laser Sint, (**D**) Laser Mach, and (**E**) Ti Blast implants. Scale bars for each sequence of microphotographs are as follows: 1 mm (top left), 500 μm (top right), 50 μm (bottom left), and 40 μm (bottom right).

**Figure 2 biomolecules-13-01048-f002:**
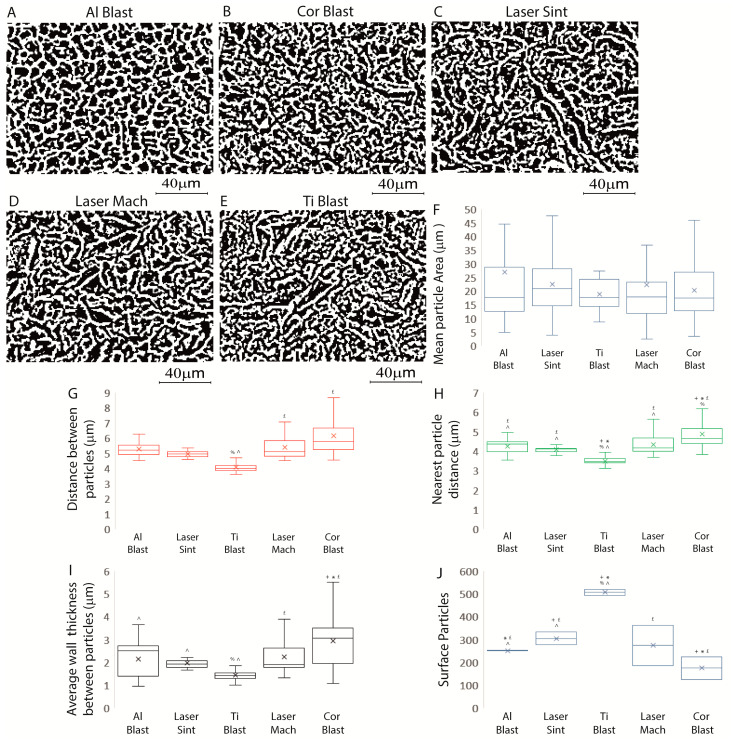
Summary of the titanium dental implants’ textures. (**A**) Al Blast, (**B**) Cor Blast, (**C**) Laser Sint, (**D**) Laser Mach, and (**E**) Ti Blast implants’ binarized and thresholded SEM microphotographs were analyzed to quantify the mean particle area (**F**), the distance between the particles (**G**), the nearest particle distance (**H**) the average wall thickness between the particle (**I**) and their surface (**J**). Results are expressed as the mean ± SD of three independent experiments, with + Al Blast, * Laser Sint, £ Ti Blast, % Laser Mach, and ^ Cor Blast indicating significance (ANOVA followed by Bonferroni’s post hoc test).

**Figure 3 biomolecules-13-01048-f003:**
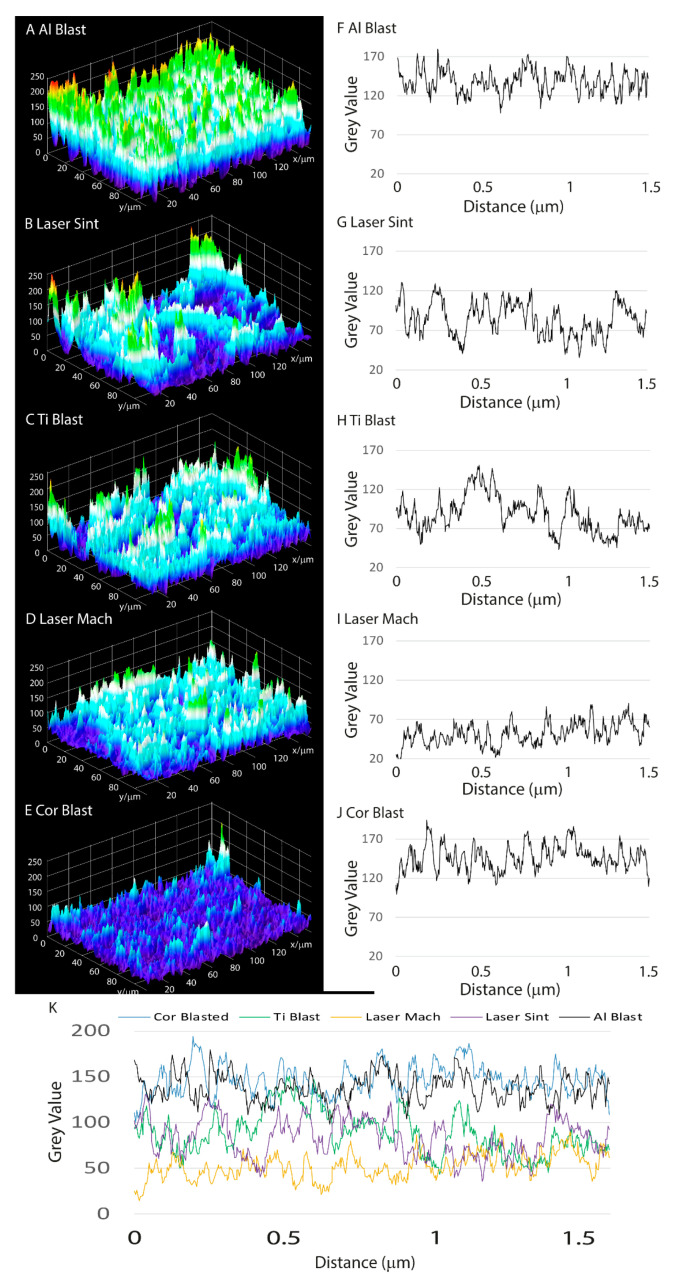
Dental implants’ interfacial roughness profile graphs and 3D profiles. 3D profiles and interfacial roughness profile graphs of (**A**,**F**) Al Blast, (**B**,**G**) Laser Sint, (**C**,**H**) Ti Blast, (**D**,**I**) Laser Mach, and (**E**,**J**) Cor Blast implants; (**K**) combined roughness profiles of Al Blast, Laser Sint, Ti Blast, Laser Mach, and Cor Blast implants.

**Figure 4 biomolecules-13-01048-f004:**
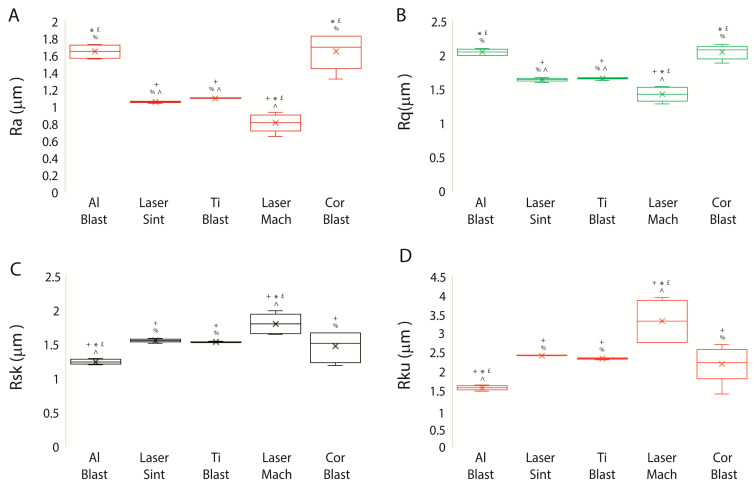
Roughness-associated parameters of dental implants. (**A**–**D**) Graphs highlighting the differences in (**A**) Ra, (**B**) Ru, (**C**) Rsk, and (**D**) Rku found among the surfaces of the implants. Results are expressed as the mean ± SD of three independent experiments, with + Al Blast, * Laser Sint, £ Ti Blast, % Laser Mach, and ^ Cor Blast (ANOVA followed by Bonferroni’s post hoc test) indicating significance. Abbreviations: arithmetical mean deviation (Ra), root mean square deviation (Rq), skewness of the assessed profile (Rsk), kurtosis of the assessed profile (Rku).

**Figure 5 biomolecules-13-01048-f005:**
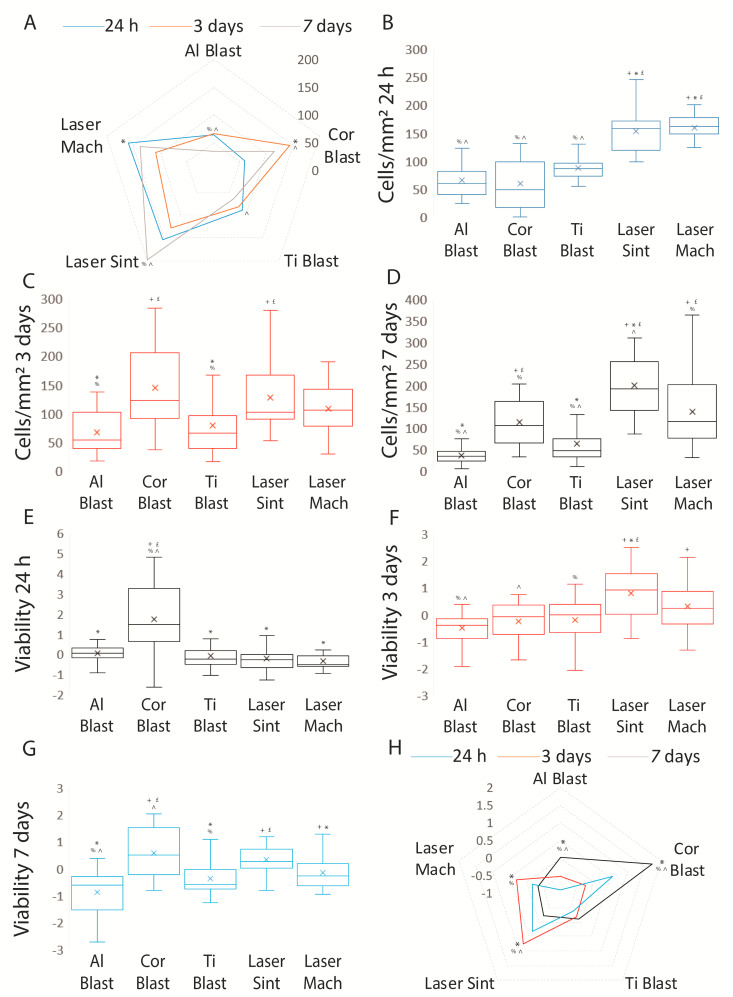
Graphs showing the cell count of ADSCs at 24 h, three days, and seven days after being cultured on the whole−body commercial implants. Graph (**A**) compares the results of adhesion and proliferation at the three different time points; graph (**B**) depicts adhesion at the 24 h time point; graph (**C**) illustrates adhesion at 3 days; graph (**D**) shows adhesion at 7 days; graph (**E**) depicts viability at the 24 h time point; graph (**F**) illustrates viability at 3 days; graph (**G**) shows viability at 7 days; and graph (**H**) compares the results of viability at the three different time points. Results are expressed as the mean ± SD of three independent experiments, with + Al Blast, * Cor Blast, £ Ti Blast, % Laser Sint, and ^ Laser Mach (ANOVA followed by Bonferroni’s post hoc test), and * T1 and 24 h–3 days, ^ T2 and 24 h–7 days, and % T3 and 3–7days (paired *t*-test) indicating significance.

**Figure 6 biomolecules-13-01048-f006:**
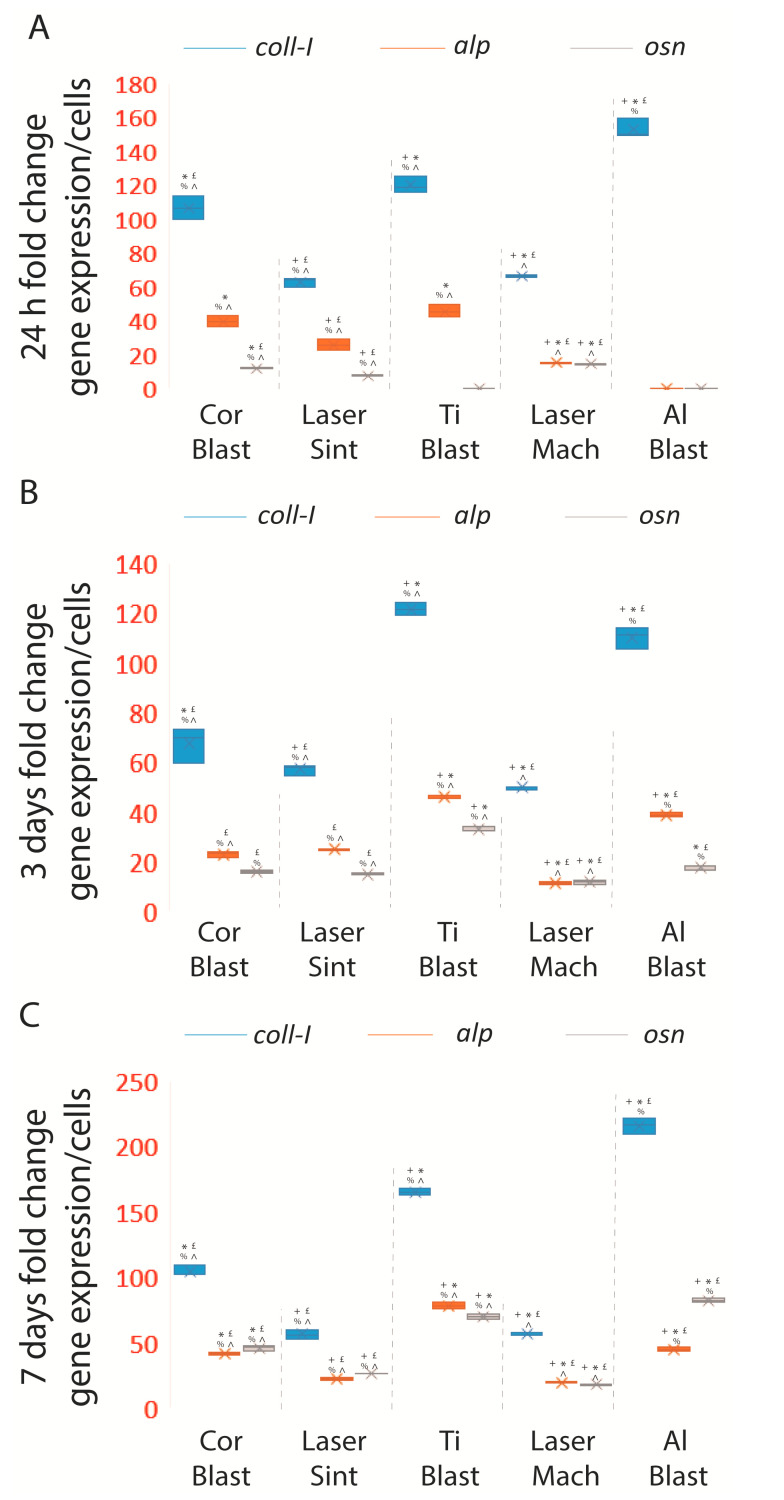
Summary of the spontaneous ADSCs differentiation resulting from their seeding on tested whole-body dental implants. After 24 h, three days, and seven days, ADSCs cultured on whole-body commercially available dental implants expressed extracellular matrix and osteoblastic markers: (**A**) 24 h, (**B**) 3 days, and (**C**) 7 days. Results are expressed as the mean ± SD of three independent experiments with + Al Blast, * Laser Sint, £ Ti Blast, % Laser Mach, and ^ Al Blast (ANOVA followed by Bonferroni’s post hoc test) denoting significance. Abbreviations: alkaline phosphatase (*alp*); osteonectin (*osn*); collagen type 1 (*coll*-*I*). At the same time point, significant *alp* expressions were achieved under the effect induced by, in decreasing order, the Ti Blast (45.6 ± 3.9 FC), Cor Blast (39.4 ± 3.1 FC), Laser Sint (25.8 ± 3.3 FC), and Laser Mach (15.27 ± 0.5 FC) implants, while the Al Blast implant showed no *alp* expression (Figure 6B; Appendix A).

**Figure 7 biomolecules-13-01048-f007:**
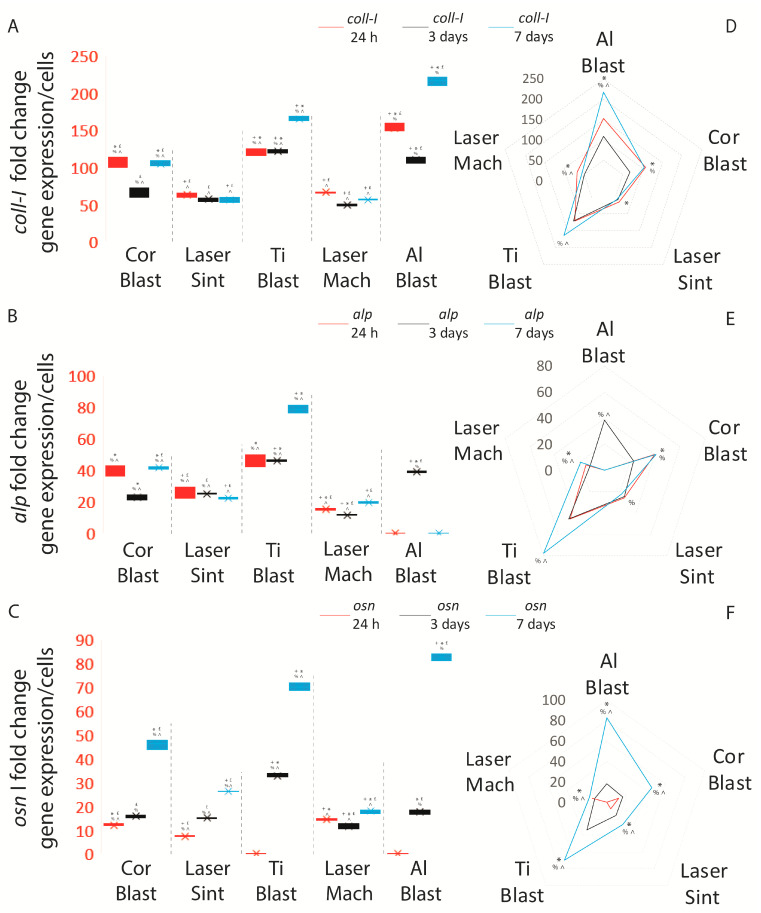
Summary of the spontaneous ADSCs differentiation resulting from their seeding on the whole-body dental implants. (**A**,**D**) *coll*-*I*, (**B**,**E**) *alp*, and (**C**,**F**) *osn* expression of ADSCs cultured on the tested dental implants after 24 h, three days, and seven days. Results are expressed as the mean ± SD of three independent experiments, with + Al Blast, * Laser Sint, £ Ti Blast, % Laser Mach, and ^ Al Blast (ANOVA followed by Bonferroni’s post hoc test), and * 24 h–3 days, ^ 24 h–7 days, and % 3–7days (paired *t*-test) indicating significance. Abbreviations: alkaline phosphatase (*alp*); osteonectin (*osn*); collagen type 1 (*coll*-*I*).

**Table 1 biomolecules-13-01048-t001:** Titanium dental implants’ surface particle characteristics. Summary of the results obtained from surface particle analysis.

	Al Blast	Cor Blast	Laser Sint	Laser Mach	Ti Blast
Particle Area	26.87 ± 29.74	20.26 ± 11.99	22.5 ± 10.87	22.25 ± 21.94	18.84 ± 7.27
Distance between Particles (µm)	5.27 ± 0.44	6.17 ± 1.17	4.95 ± 0.25	5.4 ± 0.76	4.06 ± 0.3
Nearest Particle Distance (µm)	4.26 ± 0.38	4.86 ± 0.69	4.07 ± 0.2	4.34 ± 0.49	3.5 ± 0.22
Average Wall Thickness between Particles (µm)	2.12 ± 0.76	2.93 ± 1.21	1.98 ± 0.3	2.24 ± 0.74	1.44 ± 0.25
Particles per Surface	250 ± 1.5	174 ± 69	304 ± 39	273 ± 126	506 ± 18

**Table 2 biomolecules-13-01048-t002:** Summary of the roughness-related characteristics of the dental implants under study. Summary of the roughness parameters assessed on the tested dental implants. The results of the ANOVA and Bonferroni’s post hoc test indicate that the data are significant, with * denoting *p* ≤ 0.05. Abbreviations: arithmetical mean deviation (Ra), root mean square deviation (Rq), skewness of the assessed profile (Rsk), kurtosis of the assessed profile (Rku).

	Al Blast	Cor Blast	Laser Sint	Laser Mach	Ti Blast
Ra	1.64 ± 0.07	1.64 ± 0.21	1.05 ± 0.01	0.8 ± 0.1	1.09 ± 0.004
Rq	2.04 ± 0.04	2.04 ± 0.1	1.63 ± 0.02	1.42 ± 0.1	1.65 ± 0.01
Rsk	1.23 ± 0.04	1.45 ± 0.2	1.54 ± 0.02	1.79 ± 0.15	1.52 ± 0.005
Rku	1.56 ± 0.06	2.19 ± 0.5	2.4 ± 0.007	3.31 ± 0.56	2.32 ± 0.02

**Table 3 biomolecules-13-01048-t003:** Summary of the tested implants in utilizing the 3D approach: The table summarizes the distinct results following 3D cell cultures on ten different whole-body titanium implants.

Name		Roughness	Adhesion	Viability and Proliferation	Differentiation
Ti Blast, grade 5 titanium	Lower distance between particles, nearest average wall thickness. Higher number of surface particles	Intermediate, similar to HA Blast	Lower than HA Blast	Lower than laser-treated, T2 and T3 decrease. Lower viability than Laser Sint and Cor Blast	*coll I*, *alp*, and *osn* high, but lower than HA Blast. No *osc*
Al Blast, grade 4 titanium	Distance between particles and nearest particle distance similar to HA Blast. Lower average wall thickness	Higher Ra, Rq and lower rsk, rku than HA Blast	Lower than HA Blast	Lower than laser-treated, T2 and T3 decrease. Low viability	*coll I*, *alp*, and *osn* high, but lower than HA Blast. No *osc*
Laser Sint, grade 5 titanium	Distance between particles, nearest particle distance similar to HA Blast. Lower average wall thickness and higher number of particles than HA Blast	Roughness intermediate, similar to HA Blast.	Intermediate, lower than HA Blast	Highest proliferation and viability	*coll I, alp*, and *osn* intermediate. No *osc*
Laser Mach, grade 5 titanium	Distance, number of particles, and distance between particles similar to HA Blast. nearest particle distance and average wall thickness lower	Roughness intermediate, similar to HA Blast, but lower Ra	Intermediate, lower than HA Blast	High but lower than Laser Sint, T2 decrease and T3 increase. Viability lower than Laser Sint and Cor Blast	*coll I*, *alp*, and *osn* intermediate. No *osc*
Cor Blast, grade 5 titanium	Number of particles, distance between particles similar to HA Blast. Nearest particle distance and average wall thickness lower than HA Blast	Roughness higher than HA Blast, higher Ra, Rq, and similar Rsk, Rku	Lower than HA Blast	Intermediate, T2 increase and T3 decrease. Highest viability	*coll I, alp*, and *osn* intermediate. No *osc*
Plasma Spray, grade 4 titanium	Distance between particles, nearest particle distance, and average wall thickness lower than HA Blast. Higher number of particles than HA Blast	Roughness lower than HA Blast. Lower Ra and Rq and higher Rsk, Rku	High but lower than HA Blast	Lower than laser-treated, T2 and T3 decrease. Low viability	*coll I*, *alp*, and *osn* low. No *osc*
Laser, grade 4 titanium	Distance between particles, average wall thickness similar to HA Blast. Lower nearest particle distance and higher number of particles than HA Blast.	Roughness lower than HA Blast. Lower Ra and Rq and higher Rsk, Rku	Intermediate, lower than HA Blast	Highest proliferation, T2 increase and T3 stable, good viability	*coll I*, *alp*, and *osn* intermediate. No *osc*
Double Acid etching, grade 4 titanium	Higher number of particles than HA Blast. Distance between particles, nearest particle distance, and average wall thickness lower than HA Blast.	Roughness lower than HA Blast. Low Ra and Rq and high Rsk, Rku	High, but lower than HA Blast	Lower than laser-treated, T2 decrease and T3 increase, intermediate viability	*coll I, alp*, and *osn* low. No *osc*
HA Blast “New”, grade 5 titanium	Number of particles, distance between particles, nearest particle distance, and average wall thickness similar to HA Blast	Roughness lower than HA Blast. Low Ra and Rq and high Rsk, Rku	High, but lower than HA Blast	Lower than laser-treated, T2 stable and T3 decrease. Intermediate viability	*coll I*, *alp*, and *osn* high, but lower than HA Blast. No *osc*

## Data Availability

The datasets generated and/or analyzed during the current study are available upon reasonable request from the corresponding author.

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
