# Peer review of "Considering the Value of 3D Cultures for Enhancing the Understanding of Adhesion, Proliferation, and Osteogenesis on Titanium Dental Implants"

_biomolecules, 2023, doi:10.3390/biom13071048_

Round 1

Reviewer 1 Report

NA

Author Response

Reviewer 1

We appreciate your taking the time to read our paper. In this resubmitted version, we worked hard to address all of the comments and suggestions, and the paper has been much improved as a consequence of the reviewers' remarks.

We organized this letter to make the review process easier by addressing reviewers' points (shown in red) in the order they were made.

Considering the value of 3D culture for enhancing the understanding of adhesion, proliferation, and osteogenesis on titanium dental implants Overall, it is a well-written article describing the scientific backgrounds, methodologies, supportive results, and discussion. Very few suggestions to improve some existing issues:

  • Line 128 “thanks”, I thought you meant tank?

Ans: Thank you to the reviewer; the term "thanks" has been replaced with "due" at Line 128.

  • Ti6Al4V, use the superscript

Ans: The formula Ti6Al4V (Line 151) has been changed in accordance with the recommendation at Line 147, thanks to the reviewer's kind advice.

  • Why these five specific types of implants were chosen? Adding reasons can help readers.

Ans: Thanks about the reviewer’s suggestion the specification about the criteria used to choose the implants has been added and reported as follows (Line 124) “The implants had a tronco-conical shape with dimensions ranging from 3.6-4 mm in diameter to 8-9 mm in length and were chosen for their wider market representation, which could be attributed to their higher difference in roughness degree and surface texture compared to our previous research[24]”.

  • It is not clear if your research group made those five implants or you purchased/acquired them. If you made, it needs more processing methods to add, if not cite the manufacturer as per Journal requirement.

Ans: The manufacturing companies provided us with the implants. Even though we agree with the reviewer, we chose to keep their names anonymous because not all of them granted us permission.

  • Graphs in Figure 5(A, H), 6, and 7 are not well-visible and readable. Clear Figures are needed there.

Ans: Thank you for the reviewer's feedback. The figures have been modified to make them more visible and understandable, and they have also been added to the text at 300 dpi, as follows:

Fig. 5:

Fig. 6:

Fig. 7:

Reviewer 2 Report

As stated by the authors, the aim of the present study is “to examine whether the microgeometry and surface properties of five titanium implants affect the adhesion, proliferation, and differentiation in vitro of a population of adipose-tissue derived stem cells (AD-92 SCs) in order to strengthen the groundwork for a more realistic combinatorial strategy and confirm the significance of testing cells in a 3D experimental condition

Five different titanium dental implants were tested:

1: sandblasted with aluminum oxide and acid etched

2: laser sintered

3: sandblasted with titanium dioxide spheres and fluoride treatment

4: laser-machined

5: sandblasted with corundum and different acid etchings

I have the following comments:

The title does not reflect the reported study – The authors did a 2D culture over the implants surface that were placed vertically – this is not a 3D culture !!

Nothing new is added to the state of the art. It is well known, for a long time, that the implant roughness and topography affect the adhesion, proliferation and osteoblastic differentiation of pluripotent cells from several sources. 

Although authors state that they explored a “combinatory strategy” – they did just a routine cytocompatibility evaluation on implants placed vertically. For proving a combinatory approach, further studies will be needed, namely the implantation of the colonized implants in an in vivo bone regeneration model and evaluation of the osteointegration process.

Although the study was conducted with a correct standard routinely used methodology, an essential information is missing. In cytocompatibility studies, images are mandatory – SEM images of the colonized implants are needed (low and high magnification) – they would provide information on the cell attachment, morphology, pattern of cell growth and the intimate interactions with the material surface topography, that will affect the all implant integration process.

Lastly, but very important, the same research team reported recently a very similar work (objectives, methodology, results) - “Assessing the Efficacy of Whole-Body Titanium Dental  Implant Surface Modifications in Inducing Adhesion, Proliferation, and Osteogenesis in Human Adipose Tissue Stem Cells”, Funct. Biomater. 2022, 13, 206 - that provided similar conclusions. The only difference was the tested implant surface modifications: “Five different titanium dental implants were compared in this study, as follows: plasma spray; laser; HA-blasted with hydroxyapatite (HA) and bland acid etching; double acid etching; and HA-blasted with HA followed by “new” patented treatment”.  

Author Response

Reviewer 2

First and foremost, we want to thank you all for taking the time to read our manuscript. We worked hard to address all of the comments and concerns, and the article has been considerably improved as a result of the reviewers' observations in this resubmitted version.

We arranged this letter to simplify the review process by addressing reviewers' comments point-by-point (shown in red) in the same order they were made.

As stated by the authors, the aim of the present study is “to examine whether the microgeometry and surface properties of five titanium implants affect the adhesion, proliferation, and differentiation in vitro of a population of adipose-tissue derived stem cells (AD-92 SCs) in order to strengthen the groundwork for a more realistic combinatorial strategy and confirm the significance of testing cells in a 3D experimental condition

Five different titanium dental implants were tested:

1: sandblasted with aluminum oxide and acid etched

2: laser sintered

3: sandblasted with titanium dioxide spheres and fluoride treatment

4: laser-machined

5: sandblasted with corundum and different acid etchings

I have the following comments:

The title does not reflect the reported study – The authors did a 2D culture over the implants surface that were placed vertically – this is not a 3D culture !!

Ans: A 3D cell culture, by definition, is an artificially established environment in which cells are allowed to grow and interact with their surroundings in all three dimensions. In comparison to commonly used methods (titanium disks), this is a 3D technique because it permits cells to be attracted by the implant surface and interact in all three dimensions rather than adhering just by gravity. Furthermore, unlike 2D environments (such as a Petri plate), this 3D cell culture permits cells to proliferate in all directions if given enough time, just like in vivo. Lastly, since very few 3D approaches are currently being tried to test dental implants, with this study and the previous one, we wanted to highlight and encourage researchers to understand the relevance and utility of the approach beyond the traditional 2D culture studies on titanium disks.

Nothing new is added to the state of the art. It is well known, for a long time, that the implant roughness and topography affect the adhesion, proliferation and osteoblastic differentiation of pluripotent cells from several sources. 

Ans: There are many articles in the literature claiming the importance of implant microtopography in 2D culture settings. On the contrary, only a few 3D approaches have been documented; thus, we hoped to confirm, by comparing, extrapolating, and emphasizing the results obtained from both papers, the superior potential of the 3D method and thus support its wider use to better test and favour a combinatorial approach with implants or prostheses in general.

Indeed, the results revealed that, in addition to the roughness of the implants under consideration, which were representative of a wider implant market, the 3D culture approach highlighted substantial differences, in fact:

  1. The adhesion and proliferation of the laser-treated implants exhibited the highest degree, possibly due to the influence of the post-processing methods.
  2. The sandblasted implant surfaces performed better in terms of osteoinduction, possibly via the chemical treatments the implants underwent during the manufacturing process.
  3. The Ti Blast osteoinductive capability might have benefited from the highest texture density and the fluorine surface treatment.

The findings also showed that when surface roughness and manufacturing processes are comparable or identical, the difference in cell adhesion and differentiation between HA Blasted and the other implants can be attributed to the surface treatment, and that the 3D approach was capable of assessing those differences.

Although authors state that they explored a “combinatory strategy” – they did just a routine cytocompatibility evaluation on implants placed vertically. For proving a combinatory approach, further studies will be needed, namely the implantation of the colonized implants in an in vivo bone regeneration model and evaluation of the osteointegration process.

Ans: We are not sure where the reviewer noticed that we investigated the combinatorial approach; all we claimed was that the 3D culture condition will boost the combinatorial approach's efficacy.

To recall what was discussed in the study, below are our assertions concerning the combinatorial approach: “Individuals with pathologic conditions and restorative deficiencies might benefit from a combinatorial approach encompassing stem cells and dental implants; however, due to the various surface textures and coatings, the influence of titanium dental implants on cells exhibits extensive wide variation.”, “In addition, the 3D method would allow researchers to test various implant surfaces more thoroughly. Integrating with preconditioned stem cells would inspire a more substantial combinatorial approach to promote a quicker recovery to patients with restorative impairments.”, “Indeed, recent advances in combinatorial strategies using stem cell-based tissue engineering have improved titanium implant osseointegration in diabetic and osteoporotic animal models [1,12,16].”, “According to these investigations, titanium implants' ability to osseointegrate in an healthy, diabetic and osteoporosis models is favourably impacted by this combinatorial strategy employing stem cell sheets and may expand the clinical indications in patients with osteoporosis and improve the success rate of dental implant placement [1,12,16].”, “As a consequence, the aim of the present study is to examine whether the micro-geometry and surface properties of five titanium implants affect the adhesion, proliferation, and differentiation in vitro of a population of adipose-tissue derived stem cells (ADSCs) in order to strengthen the groundwork for a more realistic combinatorial strategy and confirm the significance of testing cells in a 3D experimental condition.”  and “Finally, by implementing this 3D approach and the subsequent advancement of the combinatorial approach in conjunction with pre-clinical contexts, it would be possible to identify the most efficient and appropriate surface to be used in combination with preconditioned stem cells, promoting a more significant and fast recovery while significantly improving the healing process in individuals with restorative deficiencies.”

Although the study was conducted with a correct standard routinely used methodology, an essential information is missing. In cytocompatibility studies, images are mandatory – SEM images of the colonized implants are needed (low and high magnification) – they would provide information on the cell attachment, morphology, pattern of cell growth and the intimate interactions with the material surface topography, that will affect the all implant integration process.

Ans: We appreciate the feedback and the advice. We concur with the reviewer's assessment of the SEM image analysis following cell culture, but at the moment—due to difficulties with setting up the experiment and the impossibility of recovering new implants—we were unable to collect any images of the implants following the 3D culture.

Lastly, but very important, the same research team reported recently a very similar work (objectives, methodology, results) - “Assessing the Efficacy of Whole-Body Titanium Dental Implant Surface Modifications in Inducing Adhesion, Proliferation, and Osteogenesis in Human Adipose Tissue Stem Cells”, Funct. Biomater. 2022, 13, 206 - that provided similar conclusions. The only difference was the tested implant surface modifications: “Five different titanium dental implants were compared in this study, as follows: plasma spray; laser; HA-blasted with hydroxyapatite (HA) and bland acid etching; double acid etching; and HA-blasted with HA followed by “new” patented treatment”.  

Ans: We explicitly acknowledged previous paper's presence and included it as a reference and in the discussion. The present findings are novel, and the findings of this study improve the past research, reinforcing the assertion that this 3D approach is significantly superior and may be employed with increasing efficacy in the future.

Reviewer 3 Report

A manuscript that will contribute to the literature after correcting the many English language errors, sentence structure, and word use.

The reviewer has provided suggestions. However, the entire manuscript must be thoroughly edited.

An English language service is recommended.

Author Response

Reviewer 3

We thank you, the reviewer, for the time spent reading our article. In this resubmitted version, we tried hard to address all of the reviewer's concerns and thoughts, and the article has been substantially improved as a result of his/her comments.

Ans: The article has been carefully checked for grammatical and spelling mistakes, as kindly suggested by the reviewer. Our and the reviewer's corrections are highlighted in red throughout the text to facilitate the review process.

Amplitude and surface chemistry may also be factors. For example, adsorption of adhesion molecules from the media.

Ans: As a result of the reviewer's keen feedback, we also added a part regarding the amplitude of the surface roughness and the surface chemistry, which is reported as follows (Line 568) “). Indeed, both the amplitude of surface roughness and the surface chemistry of HA-coated implants have been shown to increase cell adhesion [22,24,53] and subsequent differentiation [22,24,53–57]. In fact, the greater wettability and hydrophilicity stimulate the ad-sorption of adhesion molecules from the media and, thereby, from the blood onto the titanium implant surface, promoting cell adhesion [53–56]. Moreover, the HA (Ca10(PO4)6(OH)2) coating’s compositional and structural similarity to bone [58] enhances stem cell differentiation [24,58] stimulating extracellular matrix protein secretion and mineralization [57] via integrin-mediated signal transduction [53,59]. On the down side, the primary drawback of using HA is its limited ability to adhere to metal substrates; however, many researchers are making efforts to improve the techniques [58,60,61]”.

Round 2

Reviewer 2 Report

Although i do not agree with the definition of a 3D culture system given by the authors, i accept their response.

Reviewer 3 Report

The authors have addressed my concerns.